

# *In vitro* performance in cotton plants with different genetic backgrounds: the case of *Gossypium hirsutum* in Mexico, and its implications for germplasm conservation

Alejandra Hernández-Terán[1,2], Ana Wegier[3], Mariana Benítez[1,4], Rafael Lira[5], Tania Gabriela Sosa Fuentes[3] and Ana E. Escalante[1]

[1] Laboratorio Nacional de Ciencias de la Sostenibilidad, Instituto de Ecología, Universidad Nacional Autónoma de México, Mexico City, Mexico
[2] Programa de Doctorado en Ciencias Biomédicas, Universidad Nacional Autónoma de México, Mexico City, Mexico
[3] Jardín Botánico, Instituto de Biología, Universidad Nacional Autónoma de México, Mexico City, Mexico
[4] Centro de Ciencias de la Complejidad, Universidad Nacional Autónoma de México, Mexico City, Mexico
[5] Facultad de Estudios Superiores Iztacala, Universidad Nacional Autónoma de México, Los Reyes, Mexico

## ABSTRACT

One of the best *ex situ* conservation strategies for wild germplasm is *in vitro* conservation of genetic banks. The success of *in vitro* conservation relies heavily on the micropropagation or performance of the species of interest. In the context of global change, crop production challenges and climate change, we face a reality of intensified crop production strategies, including genetic engineering, which can negatively impact biodiversity conservation. However, the possible consequences of transgene presence for the *in vitro* performance of populations and its implications for biodiversity conservation are poorly documented. In this study we analyzed experimental evidence of the potential effects of transgene presence on the *in vitro* performance of *Gossypium hirsutum* L. populations, representing the Mexican genetic diversity of the species, and reflect on the implications of such presence for *ex situ* genetic conservation of the natural variation of the species. We followed an experimental *in vitro* performance approach, in which we included individuals from different wild cotton populations as well as individuals from domesticated populations, in order to differentiate the effects of domestication traits dragged into the wild germplasm pool via gene flow from the effects of transgene presence. We evaluated the *in vitro* performance of five traits related to plant establishment ($N = 300$): propagation rate, leaf production rate, height increase rate, microbial growth and root development. Then we conducted statistical tests (PERMANOVA, Wilcoxon post-hoc tests, and NMDS multivariate analyses) to evaluate the differences in the *in vitro* performance of the studied populations. Although direct causality of the transgenes to observed phenotypes requires strict control of genotypes, the overall results suggest detrimental consequences for the *in vitro* culture performance of wild cotton populations in the presence of transgenes. This provides experimental, statistically sound evidence to support the implementation of transgene screening of plants to reduce time and economic costs in *in vitro* establishment, thus contributing to the overarching goal of germplasm conservation for future adaptation.

Corresponding author
Ana E. Escalante,
aescalante@iecologia.unam.mx

# INTRODUCTION

Interest in plant germplasm conservation addresses the need to preserve a diverse genetic pool, thus providing options for future decision-making (*Rockstrom et al., 2014*). Such options must include genetic and phenotypic diversity to face current and future challenges in crop production. The conservation of these variants can help in developing or finding solutions to disease, changing environments, and low yields, among others, and is necessary for safeguarding biodiversity and cultural identity (*Hawkes, 1977*; *Plucknett et al., 1983*; *Hajjar & Hodgkin, 2007*). In most crops, the largest genetic variation exists in the Crop Wild Relatives (CWR) found in the centers of origin of the species (*Hawkes, 1977*). Accordingly, research groups in food security identified CWR as a target group for conservation (*Harlan, 1965*; *Hunter & Heywood, 2011*; *Castañeda Álvarez et al., 2016*). Due to the importance of CWR for conservation, international agreements have made the *in situ* and *ex situ* preservation of their genetic diversity one of their goals (Aichi Target 13) (*Leadley et al., 2014*).

Plant tissue culture methods represent a robust approach for many purposes, from being a tool designed to pursue a variety of basic research questions to helping *ex situ* preservation of genetic diversity (*Engelmann, 1991*; *Gosal & Kang, 2012*). The most successful example of tissue culture in commercial and conservation applications is micropropagation: the propagation of plants from small parts under *in vitro* conditions. The success of micropropagation in tissue culture is due to ease of having multiple genetic clones from different geographic locations, thus lowering the risk of loss of such genotypes (*Kumar & Reddy, 2011*; *Rajasekharan & Sahijram, 2015*).

Despite the great advantages of micropropagation for *in vitro* conservation of plant species, different factors can compromise its success, such as culture medium composition, environmental conditions and genotype, among others (*Li et al., 2002*; *Tyagi et al., 2004*; *Kumar & Reddy, 2010*). In fact, it has been shown that different cultivars of the same species can have different *in vitro* performance or success (*Gubis et al., 2003*; *Pathi & Tuteja, 2013*). This reveals the sensitivity of *in vitro* culture to even small genetic variation. In the past few decades, the use of Genetically Modified Organisms (GMO) has become extensive (*ISAAA, 2017*) and a source of new genetic variation even in countries that are considered centers of origin of important crops (*Lu, 2008*). In some cases, the release of GMOs in areas with CWR or with other crops and weeds, has caused gene flow events across populations of different economically important cultivars, such as maize, cotton, papaya, bent grass, alfalfa and canola (*Quist & Chapela, 2001*; *Warwick et al., 2008*; *Piñeyro Nelson et al., 2009*; *Wegier et al., 2011*; *Greene et al., 2015*; *Manshardt et al., 2016*). Thus, given the extensive use of GMO technologies in economically important cultivars, it becomes relevant to analyze all evidence related to the effects of this introduced variation on the *in vitro* culture germplasm conservation efforts.
Mexico is the center of origin for cotton (*Gossypium hirsutum* L.) (*Ulloa et al., 2006*; *Burgeff et al., 2014*), and its metapopulations have been found on the coasts of the country, while extensive cultivars can be observed in the northern states, and backyard/home garden plants and native varieties have been reported in the southeastern states (*Velázquez-López et al., 2018*). Previous studies have described the genetic diversity of the Mexican metapopulations, including the presence of transgenes in some of them (*Wegier et al., 2011*), suggesting gene flow associated with specific transformation events (i.e., transgene introduction) from extensive cultivars to metapopulations. *Ellstrand (2018)* has posed the hypothesis that the majority of Mexican cotton metapopulations do not correspond with wild relatives of cotton (truly wild), but are instead a mix of escaped cultivar individuals that have evolved in wild conditions (weedy-wild). Nonetheless, even if the cotton metapopulations are weedy-wild relatives, they are part of the primary genetic pool (*Heywood et al., 2007*) and are thus of conservation interest due to their genetic diversity (*Ellstrand, 2018*). Phenotypic consequences of genetic flow (including transgene flow) into wild and domesticated lines in other species have been suggested in previous studies (*Hernández-Terán et al., 2017*); therefore, *in vitro* culture performance could also be affected, in principle, by genetic modification. This could have important consequences for the success of germplasm conservation strategies.

In the present study, and given the genetic diversity of *G. hirsutum* in Mexico, we analyzed experimental evidence of the effects of transgene presence on the *in vitro* performance of representative population clones of *G. hirsutum* diversity, and reflect on the implications of such effects for *ex situ* genetic conservation. For these reasons, and given that transgene presence in cotton metapopulations (hereafter wild germplasm) can be directly attributed to gene flow from domesticated populations, we included individuals from domesticated populations in our comparisons in order to differentiate the effect of domestication traits dragged into the wild germplasm pool via gene flow from the mere presence of transgenes. Thus, we hypothesize that (i) given that transgenes are directed to specific traits (e.g., defense and herbicide tolerance), which are not related to *in vitro* performance, we will not find differences in such performance between populations with and without transgenes, and (ii) the only differences to be found in the *in vitro* performance will be those associated with the domestication process, between wild and domesticated populations.

## MATERIAL AND METHODS

### Experimental design

To evaluate the potential effects of transgene presence on the *in vitro* culture performance of wild cotton plants and its consequences for germplasm conservation efforts, we conducted a systematic analysis of the performance of specific traits of an *in vitro* germplasm collection of cotton plants. We included a representative sample of the genetic diversity of wild population plants with ($W_T$) and without (W) transgenes (i.e., no isogenic lines), to test the effect of transgenes on the *in vitro* performance of metapopulation variants. Given the interest of this study for conservation strategies, we intentionally look for diversity of the genetic background of the analyzed clones, in other words population level diversity. In

addition, and as a preliminary attempt to distinguish the effects attributable to transgenes from the effects attributable to flow from domesticated or cultivar populations into wild populations, we included domesticated plants with ($D_T$) and without (D) transgenes in the experiment and analyses.

## Germplasm collection

In order to have a germplasm collection that is representative of the genetic diversity of wild cotton populations in Mexico, we collected seeds from individual plants in populations spanning its natural distribution or in metapopulations (*Wegier et al., 2011*). Ten seeds of each individual plant in the collection were germinated in prepared substrate (Peat Moss, agrolite, vermiculite (3:1:1)) and 50 g of slow-release Osmocote fertilizer (14N-14P-14K, [Scott's, Marysville, Ohio]), in a greenhouse under controlled conditions ($25 \pm 5\,°C$). Once the seedlings emerged, the apex and the first three axillary buds were explanted to start the *in vitro* culture. We also included germplasm of domesticated plant individuals from farmer's markets in Mexico City as representatives of the domesticated cotton populations.

## *In vitro* culture establishment and propagation
### Establishment

All the experimental procedures were done under the license of the Servicio Nacional de Inocuidad y Calidad Agroalimentaria (SENASICA) trough the Aviso de Utilización Confinada de OGM (folio: 007_2016). Once disinfected (Appendix S1), the axillary buds were transferred to culture tubes containing 6 ml of MS basal medium (*Murashige & Skoog, 1962*). Each tube was sealed with Parafilm M (Bemis, USA) to prevent contamination. The culture tubes were incubated in a growth room at 24 °C for a 12-hours photoperiod.

### Propagation

Propagation was initiated with individual explants that reached 8 cm height or the height of the culture tube, or when the initial culture exhausted the culture medium. Propagation consisted in removing the explants from the culture tubes, cutting each of the new axillary buds and planting them in a culture tube with fresh medium. The process was performed under sterile conditions in a laminar-flow hood (ThermoFisher, Massachusetts, USA). For more information, see Appendix S1.

## Detection of transgenes in the germplasm collection

To characterize the populations under evaluation (i.e., Wild (W) and Domesticated (D)), we looked for two constructions of lepidopteran resistance and one of herbicide tolerance (*Cry1Ab/Ac, Cry2Ab* and CP4EPSPS) in all individuals of the collection. This allowed us to detect 23 of the 33 transgenic cotton events released in Mexico (*ISAAA, 2018*). For the wild cotton populations (W and $W_T$) we carried out two independent tests to verify the presence of the genetic events: enzyme-linked immunoabsorbent assay (ELISA), and sequencing of Polymerase Chain Reaction (PCR) products. For the domesticated cotton populations (D and $D_T$), transgenic events were verified only with a PCR-sequencing assay. The ELISA tests were performed in duplicate using the following kits: Bt-Cry1Ab/1Ac ELISA Kit, Bt-Cry2A ELISA Kit, and Roundup Ready ELISA Kit (Agdia, Elkhart, Indiana, USA). The results were

read in a MultiskanFC Microplate Photometer (ThermoFisher Scientific, Massachusetts, USA). We considered a sample to be positive only when its absorbance was equal to or above three standard deviations from the average intensity of all negative controls and blank samples. In all ELISA plates a blank sample (extraction buffer), a negative, and a positive control provided in each detection kit were included. ELISA results are available as supplementary material (ELISA_results.xslx).

For the PCR assays, DNA extraction was performed in duplicate for each individual, following the DNA Miniprep CTAB method reported in *Wegier et al. (2011)*. The quality and concentration of the DNA were analyzed in a NanoDrop 2000 (ThermoFisher Scientific, Massachusetts, USA). The PCR assay was performed with the primers Cry1Ab/Ac (F 5′ACCGGTTACACTCCCATCGA 3′, R 5′CAGCACCTGGCACGAACT 3′), Cry2Ab (F 5′CAGCGGCGCCAACTCTACG 3′, R 5′TGAACGGCGATGCACCAATGTC 3′), and CP4EPSPS (F 5′GCATGCTTCACGGTGCAA 3′, R 5′TGAAGGACCGGTGGGAGAT 3′) from Eurofins Scientific (Brussels, Belgium). The assay was carried out according to the references provided in Appendix S3. Subsequently, the amplicons result of the PCR assay were verified by Sanger sequencing. The sequencing was done in the Laboratorio de Secuenciación Genómica de la Biodiversidad y de la Salud in the Instituto de Biología, UNAM. Raw sequences are available in GenBank platform (accession number MK089921 to MK089930; Appendix S5).

## Data collection

To evaluate the potential consequences of transgene presence for the *in vitro* performance of cotton populations, we measured different traits of individuals in our germplasm collection. All data analyzed is included as supplementary material (Supplementary_dataset.xlsx).

During the establishment of the *in vitro* germplasm collection we documented differences in propagation success in a period of two years that included a total of 4,377 axillary buds (corresponding to 74 individual original plants, 27 with transgenes and 47 without transgenes). From this original sample, we randomly selected 20 individuals (five wild with transgenes ($W_T$), five wild without transgenes (W), five domesticated without transgenes (D) and five domesticated with transgenes ($D_T$)), and 15 replicates per individual ($N = 300$) to evaluate *in vitro* performance of four phenotypic traits (leaf rate, height rate, microbial growth, and root development). This collection was intended to be a fair representation of the wild cotton metapopulations genetic backgrounds diversity, since it includes five out of the eight populations reported in Mexico (*Wegier et al., 2011*) plus ten individuals from the domesticated genetic background. Data for these traits were collected weekly during a four-month period. As mentioned above, all the experiments were conducted in a growth room at 24 °C under a 12-hours photoperiod. A detailed scheme of the experimental design is available in the Appendix S2.

Specifics for the evaluated traits are:

(i) **propagation rate**, calculated as the number of buds derived from a single individual every two weeks during two years of continual propagation, or the slope of the linear regression model using the *lm* function in *R* (no. buds *Vs.* time) ($W_T$ $N = 1800$, W $N = 2577$, D $N = 490$, $D_T$ $N = 633$);

(ii) **leaf rate**, or leaf production rate, calculated as the number of leaves derived from a single individual every week during four months of *in vitro* culture, or the slope of the linear regression model using the *lm* function in R (no. leaves *Vs.* time) ($W_T$ $N = 75$, W $N = 75$, D $N = 75$, $D_T = 75$);

(iii) **height rate**, or height increase rate, calculated as the height (cm) of a single individual measured every week during four months of *in vitro* culture or the slope of the linear regression model using the *lm* function in R (cm *Vs.* time) ($W_T$ $N = 75$, W $N = 75$, D $N = 75$, $D_T = 75$);

(iv) **microbial growth**, measured as observable growth of either bacterial or fungal organisms associated to plant tissue, potentially attributable to possible endophyte overgrowth ($W_T$ $N = 75$, W $N = 75$, D $N = 75$, $D_T = 75$). In accordance with *Quambusch & Winkelmann (2018)*, we only considered as possible endophytes those microorganisms growing directly on the explant, not in the culture media;

(v) **root development**, determined by the average number of days that it took for the roots to develop after *in vitro* establishment, multiplied by the number of individuals that developed roots, and divided by the total number of analyzed individuals ($W_T$ $N = 75$, W $N = 75$, D $N = 75$, $D_T = 75$).

No transformation of the dataset was carried out for further analysis.

### Data analysis

To determine if there are statistical differences among experimental groups for all the traits simultaneously, we carry out a Permutational Multivariate Analysis of Variance (PERMANOVA) (*Legendre & Anderson, 1999*) based on 1,000 permutations using *adonis* function in vegan R package (*Oksanen et al., 2018*). As a post-hoc test we carry out a Wilcoxon rank sums test in order to distinguish differences in the individual traits. This statistical approach was performed based on *Rebollar et al. (2017)*. In addition, to evaluate the potential differences between the analyzed populations, we conducted a Non-Metric Multidimensional Scaling (NMDS) with Manhattan distance in the *vegan R* package (*Oksanen et al., 2018*). We selected the NMDS due to its flexibility, which allowed us to use different types of variables and make few assumptions about the nature of the data (*Legendre & Legendre, 2012*). Given that the number of entries for "propagation rate" was higher than for the rest of the evaluated traits, we subsampled these entries with the "sample" function in R. To evaluate if the subsample dataset was representative of the full dataset, we conducted a paired sample *T*-test. All the analyses were conducted using R software (version 2.4-6) (*R Core Team, 2013*).

## RESULTS

### *In vitro* culture performance is significantly different between W and $W_T$ populations

The analysis for the *in vitro* culture performance traits in wild populations with ($W_T$) and without (W) transgenes shows statistically significant differences between populations for all traits (PERMANOVA, $F = 7.81$, $p = 0.0009$) and for three out of the five individual traits according to the Wilcoxon test ("height rate" $p = 4.95e-12$ ; "microbial growth"

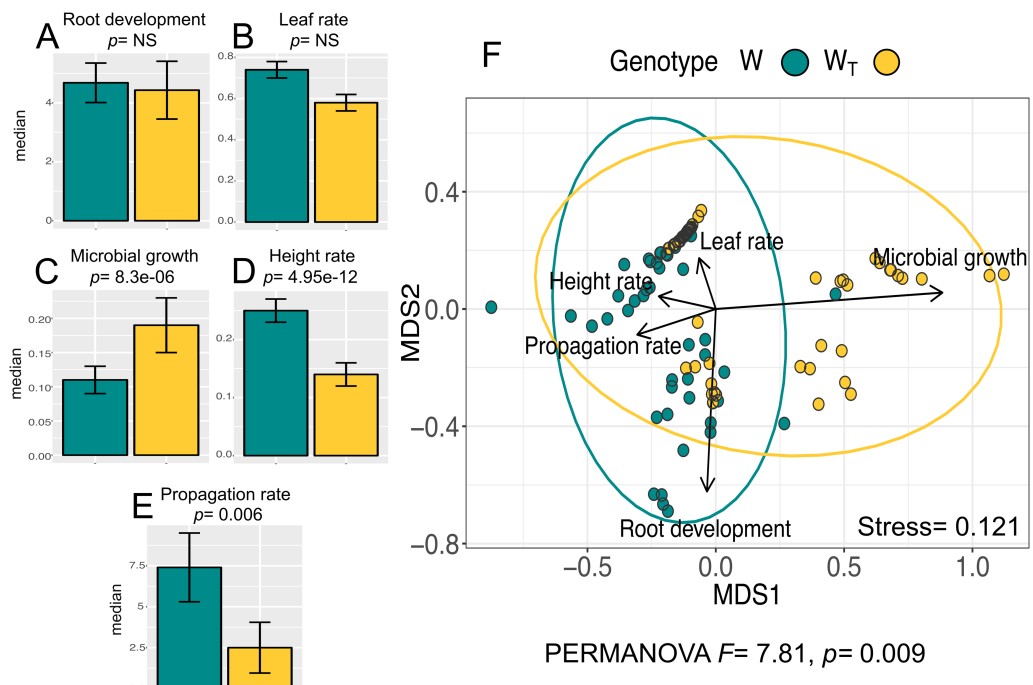

**Figure 1** *In vitro* **culture performance traits of W and W$_T$ populations.** W, wild populations without transgenes, and W$_T$, wild populations with transgenes. (A–E) median and standard error of all analyzed traits in both populations. *P* values were obtained through Wilcoxon test. (F) Non-Metric Multidimensional Scaling (NMDS) that include all the analyzed traits in the two populations. The ellipses represent 95% confidence interval around the centroids.

$p = 8.3e{-}06$; "propagation rate"; $p = 0.006$). Figure 1A–1E shows the values for all analyzed traits per population. In particular, we want to emphasize that "height rate" as an *in vitro* performance trait had the largest difference between W and W$_T$ populations.

Multivariate analysis of *in vitro* performance traits (NMDS) in W and W$_T$ populations (Fig. 1F) shows different phenotypic variations attributable to each population (W and W$_T$). "Propagation rate" is a trait positively related to W population; in contrast, "microbial growth" is positively related to W$_T$ population.

### *In vitro* culture performance differs between W and D populations

The analysis for the *in vitro* culture performance traits in wild (W) and domesticated (D) populations does show statistically significant differences between populations for all traits (PERMANOVA, $F = 9.43$, $p = 0.0009$), and for four out of the five individual traits: "leaf rate" (Wilcoxon, $p = 9.74e{-}05$), "propagation rate" (Wilcoxon, $p = 0.002$), "root development" (Wilcoxon $p = 0.001$), and "height rate" (Wilcoxon $p = 2.52e{-}11$) (Figs. 2A–2E).

Moreover, although the graphic representation of multivariate analysis of *in vitro* performance traits (NMDS) in W and D populations shows overlapping of the ellipses representing each population, the statistical analysis of multivariate differences (PERMANOVA) in both populations is statistically significant (Fig. 2F).

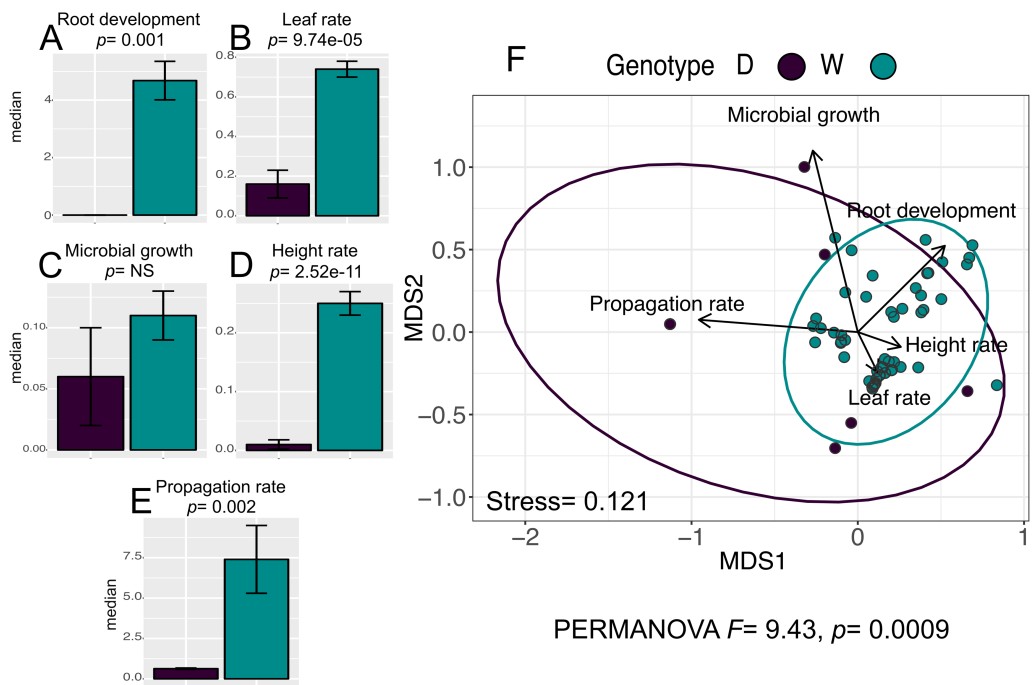

**Figure 2** *In vitro* **culture performance traits of W and D populations.** D, domesticated populations without transgenes, and W, wild populations without transgenes. (A–E) median and standard error of all analyzed traits in both populations. *P* values were obtained through Wilcoxon test. (F) Non-Metric Multi-dimensional Scaling that include all the analyzed traits in the two populations. The ellipses represent 95% confidence interval around the centroids.

### *In vitro* culture performance differs between $W_T$ and $D_T$ populations and between D and $D_T$ populations

The analysis for *in vitro* culture performance traits in wild ($W_T$) and domesticated ($D_T$) populations with transgenes shows statistically significant differences between populations in four out of the five *in vitro* performance traits (Wilcoxon "height rate" $p = 0.008$; "leaf rate" $p = 4.47e−05$; "propagation rate" $p = 0.02$; "root development" $p = 0.02$) (PERMANOVA, $F = 5.62$, $p = 0.002$). Figures 3A–3E shows the values for all the analyzed traits per population. In particular, we want to emphasize that although $W_T$ has higher values for "root development", "leaf rate" and "height rate" traits, $D_T$ population has higher "propagation rate".

In the case of the analysis of domesticated populations with ($D_T$) and without (D) transgenes, we also found statistically significant differences in three out of the five analyzed traits (Wilcoxon "microbial growth" $p = 0.001$; "propagation rate" $p = 0.03$; "root development" $p = 0.04$) (PERMANOVA, $F = 3.86$, $p = 0.0008$) (Figs. 3G–3K), showing that $D_T$ in general has a better *in vitro* performance.

The multivariate analysis of *in vitro* performance traits (NMDS) in $W_T$ and $D_T$ populations (Fig. 3F) shows different phenotypic variations attributable to each population ($W_T$ and $D_T$), which coincides with the same analysis for W and D populations (with no

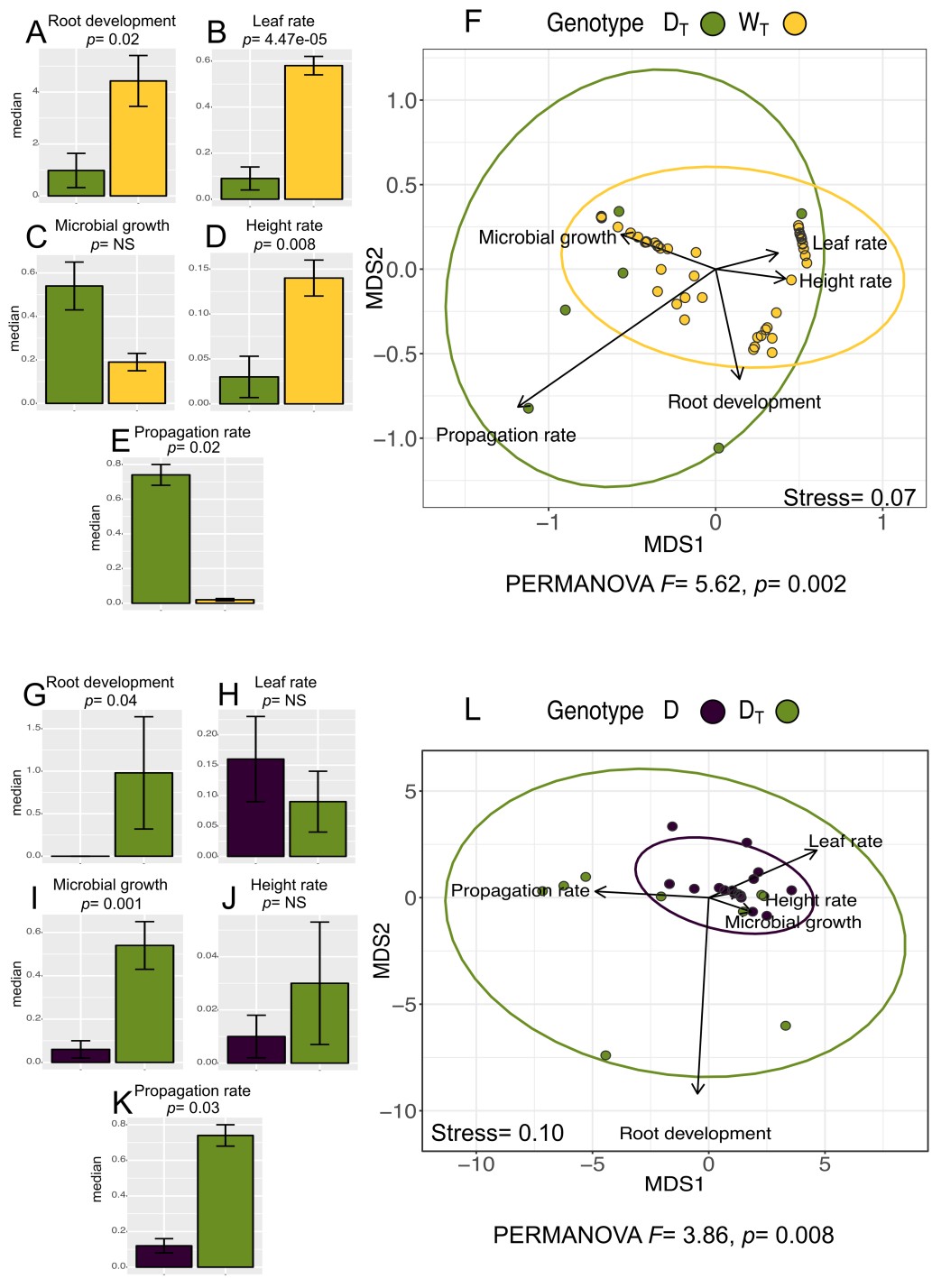

**Figure 3** *In vitro* **culture performance traits of D_T and W_T and between D and D_T populations.** In genotype, $D_T$, domesticated populations with transgenes, $W_T$, wild populations with transgenes, D: domesticated populations without transgenes. (A–E) (G–K) median and standard error of all analyzed traits in both populations. *P* values were obtained through Wilcoxon test. (F) (L) Non-Metric Multidimensional Scaling that include all the analyzed traits in the analyzed populations. The ellipses represent 95% confidence interval around the centroids.

transgenes identified; Fig. 2F). Moreover, "propagation rate" is positively related to $D_T$ population, while the rest of the traits seems positively related to $W_T$ population.

In the analysis of D and $D_T$ populations, the NMDS (Fig. 3L) shows overlapping of the ellipses representing the data set distribution of both populations, despite the significance of the statistical analysis (PERMANOVA).

To look at the full data set (W, $W_T$, D, $D_T$), and beyond the pairwise hypotheses, we carried out the NMDS and PERMANOVA analyses. The results of the full data set analyses are coherent with the pairwise comparisons, which supports the above mentioned observations (Fig. S5).

## DISCUSSION

Conservation of the genetic and phenotypic diversity in the CWR has been acknowledged as key in the preservation of diverse gene pools to secure genetic alternatives for future decisions and interventions regarding crop production. Of the different conservation strategies that exist, *in vitro* gene banks represent a robust approach to preserve genetic diversity without introducing unintended variations into the genetic pool (*Engelmann, 1991*; *Gosal & Kang, 2012*). In recent decades, the use of GMOs has become extensive (*ISAAA, 2017*), and a source of new genetic variation, even in centers of origin of important crops (*Lu, 2008*). This makes it important to evaluate evidence of the effects, if there are any, of this introduced variation on *in vitro* culture germplasm conservation efforts.

In order to analyze evidence of potential effects of transgene presence in cotton metapopulations, we compared *in vitro* performance traits in metapopulations with ($W_T$) and without (W) transgenes. We found significant differences in *in vitro* culture performance between W and $W_T$ populations for three out of the five analyzed traits (Figs. 1A–1E). Previous studies with different crop populations have shown that even small genotypic changes can have major impact in phenotypes and fitness traits both in field experimental settings (*Hernández-Terán et al., 2017*) and in *in vitro* culture (*Gandonou et al., 2005*; *Landi & Mezzetti, 2006*; *Kumar & Reddy, 2011*). Thus, it can be argued that the observed differences in *in vitro* culture performance could be the result of natural genetic variation within and among populations, however when we look at potential differences among W metapopulations, we found no statistically significant differences (Appendix S4). Nonetheless, the observed differences between $W_T$ and W populations could be attributed, as in other studies, to pleiotropic effects, where certain phenotypic traits may be linked to and affected by the genetic modification of another trait (*Filipecki & Malepszy, 2006*). In this sense, some studies have shown that genetic modification can alter metabolic pathways due to position effects of transgenes or somaclonal variation during tissue culture (*Agapito-Tenfen et al., 2013*; *Mesnage et al., 2016*). In the specific case of cotton (*Wang et al., 2015*), some authors have found overexpression of metabolites related to energy metabolism pathways that could indicate an increased demand for energy, and concomitant changes in resource allocation and development. Pleiotropic effects, have also been attributed to bottlenecks, selective sweeps, phenotypic plasticity or gene x environment (GxE) interactions (*Remington et al., 2001*; *Pozzi et al., 2004*; *Gunasekera et*
*al., 2006*; *Doust et al., 2014*). In addition, ecological costs in different plant species have been associated with the expression of transgenes (*Chen et al., 2006*). Specifically, some studies show that the physiological production of transgene toxins (e.g., *Cry* proteins) is extremely costly, limiting the energy destined for growth and reproduction. This trade-off caused by the genetic modification has been found in *Brassica* (*Snow, Andersen & Jorgensen, 1999*) and *Arabidopsis* (*Bergelson et al., 1996*). Although in our study we cannot attribute these performance differences only to the presence of transgenes (i.e., no strict control of genotypes), we can say that all individuals identified as $W_T$ were positive for the expression of transgene proteins in the ELISA approach, which is suggestive of a potential cost that gets reflected in the *in vitro* performance of $W_T$ populations. In general, *in vitro* performance differences between W and $W_T$ could be explained, at least with the information here collected, by ecological costs associated to the expression of transgenes and by potential pleiotropic and GxE interactions associated with small genetic differences.

In order to isolate the effect of domestication from the effect of transgenes on the *in vitro* performance of cotton populations, we compared the *in vitro* performance of both wild metapopulations and domesticated populations with and without transgenes. In the case of comparisons of populations without transgenes (W-D) we found significant differences in four out of the five analyzed traits (Figs. 2A–2E). Such phenotypic differentiation, regardless of substantial evolutionary divergence and genetic differentiation (*Fang et al., 2017*), is somehow unexpected, since differentiation between these populations is the result of a selective process focused on traits that are not related to *in vitro* performance, such as length, size and color of the fiber, loss of seed dispersal and germination speed (*Lubbers & Chee, 2009*; *Gross & Strasburg, 2010*; *Velázquez-López et al., 2018*). Nonetheless, strong selective forces associated with domestication and divergence times between populations are together of sufficient strength to show phenotypic differentiation even in an environment to which both W and D populations were naïve to (*in vitro* conditions).

Regarding the comparison of wild ($W_T$) and domesticated ($D_T$) populations with transgenes, we found significant differences in four out of five analyzed traits, with $W_T$ populations being better performers than $D_T$ populations in the *in vitro* culture in general (Fig. 3), with the important exception of one trait, "propagation rate". This better performance of $D_T$ populations for "propagation rate" could be the result of a history of selection in *in vitro* culture, which is part of the conventional process of genetic engineering, through which transgenes are introduced in the domesticated plants' genetic backgrounds (*Hooykaas & Schilperoort, 1992*). This suggests that since GMOs have previously gone through an *in vitro* process, it could be possible that GM plants that have been selected for culture are better adapted to these conditions. In contrast, for the rest of traits, $W_T$ populations perform better (higher "height rate", "leaf rate", and "root development"), to which a potential mechanistic explanations or hypotheses are hard to articulate. One possibility is that, given the reduced genetic variation of D populations in comparison with W populations, the general performance of D populations might be expected to be worse in environmentally astringent conditions (*Flint-garcia, 2013*; *Lu, 2013*), such as *in vitro* culture.

Regardless of the trait performance direction of populations ($W_T$ and $D_T$), we can argue that given the existent differences between D and W genotypes without transgenes (see above and *Velázquez-López et al., 2018*), it is expected that additional genetic changes (due to gene insertion) could contribute to increased phenotypic differentiation. Nonetheless, given that we did not determine the exact location of the inserted transgenes in the genome, it is not possible to give a mechanistic explanation to the specific trait differences. Overall, we can conclude that our results suggest that the presence of transgenes, originally associated with domesticated populations, has a significant impact on the *in vitro* performance of the genotypes, regardless of their wild or domesticated origin.

### Implications of transgene presence for *In vitro* wild germplasm conservation

One of the best *ex situ* conservation strategies for wild germplasm are *in vitro* banks (*Gosal & Kang, 2012*). *In vitro* conservation success depends on efficient and reliable micropropagation or *in vitro* performance of the species of interest (*Mycock, Blakeway & Watt, 2004*). Despite the reality of crop intensification, including genetic engineering, the possible consequences of the presence of transgenes for the *in vitro* performance of populations are poorly documented. In this study, we present results that suggest detrimental consequences for the *in vitro* culture performance of wild cotton populations in the presence of transgenes, which calls for monitoring transgenes in the plants to be micropropagated for conservation or future genetic improvement, as has been suggested by other authors, as conservation strategies and protocols (*Bhatia, 2015*). Moreover, it is worth noting that in the present study, with a minimal investment of three primer sets for transgene detection, we were able to identify 23 out of 33 transformation events reported for cotton populations in Mexico (*ISAAA, 2018*). As it stands, our results provide experimental evidence to support the implementation of transgene screening of plants to reduce time and economic costs in *in vitro* establishment, helping the overarching goal of germplasm conservation for future adaptation.

In current scenarios of global change, uncertain future conditions pose the major challenge of securing resources for future adaptations (*Wise et al., 2014*). In this sense, it is of utmost importance to preserve options for future decisions and to guarantee the right to biodiversity and cultural identity for future generations, which includes genetic and phenotypic options (*Rockstrom et al., 2014*). Crop biodiversity preservation is, in other words, part of our life insurance for future adaptation in a changing planet. In this sense, future work on conservation strategies and policies should put effort in expanding the knowledge about the consequences of transgene presence (*Lu, 2013*) beyond the immediate gene pool of wild populations. This means extending the efforts breadth towards other interfertile species; in other words, to the genetic primary pool.

## CONCLUSIONS

The results presented show how transgene presence in CWR cotton populations has negative consequences for their *in vitro* culture performance. In particular, reviewing our hypotheses, we found that (1) *in vitro* culture performance is significantly different between

W and $W_T$ populations, and (2) *in vitro* culture performance is different between wild and domesticated populations regardless of transgene presence. Overall, our results suggest that the presence of transgenes in wild populations imposes a cost (e.g., metabolic cost of maintaining the expression of toxins) that is reflected in their *in vitro* performance and that could endanger the success of germplasm conservation efforts. Further studies controlling for specific genotypes and specific transgene constructions would help to better disentangle the costs associated with specific genomic contexts and genetic modifications to improve genetic screenings for *in vitro* banks.

## ACKNOWLEDGEMENTS

The authors would like to thank Morena Avitia, Luis Barba Escoto, and Joel Reyna for technical assistance. The authors acknowledge Dra. Florencia García-Campusano for her support in the laboratory work, and to the Laboratorio de Biotecnología CENID-COMEF, INIFAP.

### Funding

This work was financially supported by the project "Program for the conservation of wild populations of Gossypium hirsutum in Mexico" DGAP003/WN003/18 Dirección General del Sector Primario y Recursos Naturales Renovables (DGSPRNR) that belongs to the SEMARNAT and CONABIO and the project UNAM-PAPIIT No. IN214719. Alejandra Hernández-Terán is a doctoral student from Programa de Doctorado en Ciencias Biomédicas, Universidad Nacional Autónoma de México (UNAM) and was supported by CONACYT (scholarship no. 66025). The funders had no role in study design, data collection and analysis, decision to publish, or preparation of the manuscript.

### Grant Disclosures

The following grant information was disclosed by the authors:
Program for the conservation of wild populations of Gossypium hirsutum in Mexico: DGAP003/WN003/18.
UNAM-PAPIIT: IN214719.
CONACYT: 66025.

### Competing Interests

The authors declare there are no competing interests.

### Author Contributions

- Alejandra Hernández-Terán conceived and designed the experiments, performed the experiments, analyzed the data, prepared figures and/or tables, authored or reviewed drafts of the paper, approved the final draft.
- Ana Wegier conceived and designed the experiments, contributed reagents/materials/-analysis tools, authored or reviewed drafts of the paper, approved the final draft.

- Mariana Benítez and Rafael Lira conceived and designed the experiments, authored or reviewed drafts of the paper, approved the final draft.
- Tania Gabriela Sosa Fuentes performed the experiments, approved the final draft.
- Ana E. Escalante conceived and designed the experiments, analyzed the data, contributed reagents/materials/analysis tools, authored or reviewed drafts of the paper, approved the final draft.

## DNA Deposition

The following information was supplied regarding the deposition of DNA sequences:

Data is available at GenBank, accession number MK089921 to MK089930, and in the Appendix S5.

## Data Availability

The raw data measurements are available in the Supplemental Files.

## Supplemental Information

Supplemental information for this article can be found online at http://dx.doi.org/10.7717/peerj.7017#supplemental-information.

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
