# Peer review of "In vitro performance in cotton plants with different genetic backgrounds: the case of Gossypium hirsutum in Mexico, and its implications for germplasm conservation"

_PeerJ, doi:10.7717/peerj.7017_

## Round 0.1 · original submission · Major Revisions

Please respond to the comments from the reviewers.

Please prepare new figures that have the unit of measurement of each trait. You cannot mix units of centimeters with units of microbial growth or number of leaves. Prepare one figure for each trait, containing four columns of data from W, Wt, D and Dt.

·

Basic reporting

There are some sentences that need to be revised by an expert in English editing and academic writing.

Experimental design

There are some gaps in the methodology. Though the tests implemented to prove the statistical significance of data are right, it is desirable to include the reasons explaining why such models and analyses carried our were used, including examples on the topic recently published.

Validity of the findings

Results are very well supported. Nevertheless, authors may respond some basic question such as: Why did the Wt and Dt display such phenotypes? Which metabolic pathways were changed? ...

Additional comments

The manuscript presents interesting data and fulfills aims and scope of the Peer Journal. Some weaknesses detected and that need to be addressed by authors are:
Line … 81 … “is a plant that has Mexico as its center of origin” … please revise if this sentence is appropriate and grammatically correct
Line 126 … “and 50g” … number and unit need to be separated
Line 126 … “controlled release fertilizer Osmocote), … please provide more details on this fertilizer: chemical composition, manufacturer, country of origin …
Line 134 … “6ml of MS” … number and unit need to be separated
Line 137 … “reached 8cm height” … number and unit need to be separated
Line 141 … “EUA).” … Do you mean USA? If so, please correct
Line 152 … “EUA).” Do you mean USA? If so, please correct
Line 156 … “a negative, and a positive control” … Please provide more details regarding the controls
Line 161 “Cry1Ab/Ac, Cry2Ab,” please provide the sequences of the primers
Line 162 … “CP4EPSPS” … please provide the sequence of the primer
Line 162 … “Eurofins Scientific” … include city and country of the company
Line 166 … “accession #MK089921-30” was not found in the GenBank platform:

Line 194 … “microbial growth” … please explain how you discriminated between contaminations during the preparation of culture and contaminations due to endophytes
Line 204 … “Data analysis” please include some references supporting the statistical tests you applied to analyze your data; other examples applying such tests would be desirable
Line 208 … “R” in italics, to be consistent with R in line 217
Line 227 … “populations,” comma is not needed inhere
Literature cited:
Cross reference all of the citations in the text with the references in the reference section. Make sure that all references have a corresponding citation within the text and vice versa. Double check the spelling of the author names and dates, and make sure they are correct and consistent with the citations.
Spell out all abbreviations in the text the first time they are mentioned in the text.
Spell out all journal titles in the reference section.
Make sure that all figures and tables are cited within the text and that they are cited in consecutive order.
All and all, strictly follow the instruction for authors of the journal: https://peerj.com/submissions/33125/reviews/414730/guidance/
Supplementary Material … “1g per 1lt” … please fix all units and use the SI nomenclature throughout the documents

Reviewer 2 ·

Basic reporting

The manuscript is acepted in English language. Authors give sufficient references to describe the field of this study. The hypothesis about effective of transgene presence on cotton biodiversity is an good, but the results can not robust support it.

Experimental design

no comment

Validity of the findings

no comment

Additional comments

This manuscript designed the experiment to evaluate effective of transgene presence on cotton biodiversity change. Authors examined the differences of five in vitro culture performance traits between W and D, Wt and W, WT and D, WT and DT through PERMANOVA and NMDS tests. However, the results of statistic tests are simple, and can partially reveal the effective of transgene performance on the G. hirsuturm and wild diversities. Thus other analyses must be added to in detail evaluate the effective.
Some comments followed below:
1. Based on the statistic tests of ANOVA, authors add multiple comparison tests among individuals of W, WT, D, and DT groups.
2. The title is unfit for the results, please revise it.
3. The running title is missing.
4. Line 99, Gossypium hirsutum should be omitted into G. hirsutum for the second presence in main text.
5. Line 246 and line 253 and so on, “three out of the five in vitro performance traits ” and “four out of the five analyzed traits” and so on, what are these different in vitro performance traits? Reviewer could not identify them only from Figures.
6. In Figure 2a, microbial growth comparison is wrong in label line.

Reviewer 3 ·

Basic reporting

I suggest the article to be revised by a native English speaking editor to improve the scientific vocabulary and grammatical format used to write.

Experimental design

They use population controls of domestic transgenic and wild type plants that are non-related. Therefore, I recommend using same genetic background samples in experiments.

On the other hand, the authors do not consider position insert of the genome transgene, since a gene doesn’t characterize a specific phenotype. Moreover, there are possible implications on genome variants (SNP) over populations and the possible implications associated to in vitro culture.

As well, it is necessary to determinate the endophytic bacteria or fungus that grew to correctly interpret the results. Research suggests the existence of advantages and disadvantages of the interaction between cotton plants and microorganisms, where there might be different results in in vitro culture.

Validity of the findings

The aim of the article is to evaluate in vitro culture advantages and disadvantages of transgenic and non transgenic cotton populations. However, the results are speculative due to the methods being control lacking.

Additional comments

Because of these reasons, I recommend the results given to be improved before the article is ready to be issued.

---

## Round 0.2 · Minor Revisions

Dear author

I can read that you have addressed most of the reviewers concern. The title has changed. The reviewers comments have been responded adequately and your manuscript has improved through language edititing.

Nevertheless, statistics is still a concern, and most importantly, you did not follow my editorial recommendation about the figures.

Normalizing requires the calculation of standard deviation. Dividing through the maximal value is like a percentage. Why transform the data? Why not show all the data?

I insist on my previous request:

Please prepare new figures that have the unit of measurement of each trait. You cannot mix units of centimeters with units of microbial growth or number of leaves. Prepare one figure for each trait, containing four columns of data from W, Wt, D and Dt.

De-transform your data and show the data with the original units and scale of measurements. Too much transformation and normalization generates the suspicion that the experimental data is not robust and there is something hidden. For publication in PeerJ, statistical rigor and data reliability are maximum priorities.

---

## Round 0.3 · Minor Revisions

Dear author,
Your paper has been assessed by two reviewers and myself as academic Editor.

As you can see below, the manuscript has improved considerably, but the reviewers have not yet given green light of full acceptance, but still have some minor comments. I think you can easily and rapidly incorporate them.

Please address the minor issues and submit a revised version of the manuscript. Please include a detailed response to each reviewer.

Reviewer 2 ·

Basic reporting

no comment

Experimental design

no comment

Validity of the findings

no comment

Additional comments

This revised manuscript has been improved in quality. And the explanation and information have been provided as well as the modification of statistics analysis. Now, the manuscript is clearly stated and well organized. There are two questions to be considered in revising this manuscript.
1. The microoganisms growing the explants are not always endophyte. Thus, authors should give “possible” term to state the performance.
2. The ELISA data seem to be missing, which should be added in supplemental file.

In fact, the integrated analysis of W, WT, D and DT may be better than by-pairs analysis.

Reviewer 3 ·

Basic reporting

I read your new version of the article with great interest and noticed the more adequate title, however, I have two small comments:
1.- I consider the use of superlatives in the abstract to be unusual, therefore I suggest the elimination on line 41 of the word “strongly”
2.- On line 88; I suggest “some authors” to be changed to “an author”, or add other arguments here, because they cited a Hypothesis and Theory Article, lacking any experiments, since I take this information with reserves. I invite the authors to change arguments or add other citations.

Experimental design

No comment

Validity of the findings

No comment

Additional comments

I invite authors to discuss more closely and explain in what way the presence of these transgenic elements cause a detriment to germplasm and its in vitro culture.

---

## Round 0.4 · accepted · Accept

Dear authors

I can read that you have addressed all the minor corrections.
The Manuscript has been improved.

I congratulate you for the nice piece of work, which will add value to PeerJ.

#